# Climatic niche properties shape treefrog diversity

**Felipe A. Toro-Cardona**[1,2], **Julián A. Velasco**[3], **Jesús Pinto-Ledezma**[4],
**Sean M. Rovito**[5], **Fabricio Villalobos**[6], **Octavio R. Rojas-Soto**[2], **Juan L. Parra**[1]*

**1** Grupo de Ecología y Evolución de Vertebrados, Instituto de Biología, Universidad de Antioquia, Medellín, Colombia, **2** Laboratorio de Bioclimatología, Red de Biología Evolutiva, Instituto de Ecología, Xalapa, México, **3** Instituto de Ciencias de la Atmósfera y Cambio Climático, Universidad Nacional Autónoma de México, Ciudad de México, México, **4** Department of Ecology, Evolution, and Behavior, University of Minnesota, Saint Paul, Minnesota, United States of America, **5** Unidad de Genómica Avanzada, Centro de Investigación y de Estudios Avanzados del Instituto Politécnico Nacional, Irapuato, México, **6** Laboratorio de Macroecología Evolutiva, Red de Biología Evolutiva, Instituto de Ecología, Xalapa, México

☯ These authors contributed equally to this work.
* juanl.parra@udea.edu.co

## Abstract

A suite of ecological hypotheses has been proposed to explain why more species are found in the tropics. Most of these hypotheses consider the geography of climate as a major determinant of richness, while ignoring the relationships between organisms and climate, in other words, their climatic niche. In this study, we evaluate three hypotheses that link species richness to niche properties. The niche breadth hypothesis predicts higher richness where species have narrower niches; the niche marginality hypothesis predicts higher richness where species have less marginal niches; and the niche position hypothesis predicts higher richness where species have niches similar to those of their ancestors. We estimated niche properties for 441 (70%) treefrog species from the Americas using both univariate and multivariate approaches based on minimum volume ellipsoids and projected them onto geography to relate them with the geographical pattern of species richness. We used an assemblage-based approximation to map niche properties under both approaches and performed simultaneous autoregressive models to test our hypotheses. We found support for the niche-breadth hypothesis under both methodological approaches, for the niche position hypothesis under the multivariate approach, and only for precipitation position under the univariate approach. Contrary to expectations under the niche position hypothesis for temperature, we found that treefrog species richness increased with distance to the ancestral niche temperature. We found no support for the niche marginality hypothesis under either approach. In general, our results indicate that places with high richness contain species with narrower niches, closer to their ancestral niche, but far from the mean conditions available in the Americas. These results support the long-standing hypotheses of niche packing and niche

**Data availability statement:** All data and results files are available from the figshare repository (https://doi.org/10.6084/m9.figshare.29367170).

**Funding:** Felipe A. Toro-Cardona received funding from the Posgraduate Direction at Universidad de Antioquia for international travel. the funders had no role in study design, data collection and analysis, decision to publish, or preparation of the manuscript.

**Competing interests:** The authors have declared that no competing interests exist.

conservatism, while suggesting that some niche dimensions are more constrained than others.

## Introduction

One of the major goals in geographical ecology is to identify potential mechanisms underlying current patterns in the distribution of biodiversity [1]. The decline in species richness from the tropics toward the poles (the latitudinal diversity gradient, LDG) is one of the most studied patterns in ecology and biogeography, and numerous hypotheses have been proposed to explain it [2–4]. These hypotheses can be divided into equilibrium (ecological) or non-equilibrium (historical) according to the most plausible explanation for the LDG [1]. Many of these hypotheses involve climate, time, area, and diversification rates, but few incorporate information about how organisms interact with climate, i.e., their climatic niches [2]. Although niche breadth has been proposed to account for species richness patterns, other niche properties, such as marginality and position, have been less used [5,6]. Most approaches involve a single environmental variable, usually temperature, assuming that it is the single most important aspect of climate, or ignoring other climatic variables that might be as or more important such as precipitation or seasonality [7,8]. Multivariate approaches that incorporate different niche dimensions and niche properties simultaneously are a promising avenue to explain richness patterns that have been seldom studied [9]. We hypothesize that climatic niche properties across species explain patterns of species richness at broad spatial scales.

Among the climatic niche properties, niche breadth is a proxy for species' environmental tolerances [10]. It is associated with the degree of climate specialization, where narrower niches correspond to species with a higher degree of specificity to ecological conditions [11]. Niche breadth impacts the number of co-occurring species through environmental filters that can affect species' distribution patterns [11–13] and speciation rates [14]. Tropical areas with high richness contain species with narrower niches than those in temperate zones with much lower diversity [15–17]. This relationship has been supported by multiple studies [6,18,19].

Niche marginality measures the distance between the species' niche centroid and the average climatic conditions in the region where a given clade occurs [20]. Average climatic conditions should represent those climates that are more prevalent in geography [21]. Following the species-area hypothesis [22], we expect that species richness should be higher in those climates [23]. Thus, the niche marginality hypothesis posits that species with low niche marginality (whose niche centroid is close to the average climatic conditions) should coincide with high richness areas with environments that are more prevalent, whereas species with high niche marginality should be concentrated in species-poor areas. The relationship between marginality and richness becomes intuitive when we think of environments in terms of the area that they occupy [24].

Niche position refers to the environmental values where a species' niche resides in multidimensional space [10,25]. According to the phylogenetic niche conservatism

hypothesis [26,27], species tend to track their ancestral ecological preferences over time rather than adapt to new conditions. Consequently, lineages within a clade will tend to maintain niche positions similar to those of their ancestors, limiting colonization of new environments (e.g., from the tropics to temperate areas) and promoting a higher accumulation of species in the region of origin. We consider phylogenetic niche position as the environmental distance from a species' niche centroid to the centroid of the most common ancestor of the clade [28]. We predict that areas with high species richness, where diversification is expected to be higher, should contain species with niches close to that of the most recent common ancestor (i.e., low niche position) [29]. In contrast, areas with low richness should exhibit more divergent niche positions relative to the ancestral niche. Assuming that the study group (i.e., Anurans) has a tropical origin, we would expect richness to have accumulated in areas with tropical environments.

Some studies have estimated niche properties considering temperature and precipitation as isolated dimensions of a species' environmental niche [18,30]. Other studies have used ordination methods such as principal components analysis to reduce the dimensionality of multivariate datasets and estimate niche properties like niche breadth and marginality [31], but each ordination axis is used independently. However, recent multivariate approaches to estimate niches, such as hypervolume methods, allow researchers to analyze them as n-dimensional space [32] and are more aligned with Hutchinson's niche definition [10]. The hypervolume approach may lead to differences in estimates of niche properties [33], potentially affecting relationships between species richness and niche properties.

Anurans are a suitable group for assessing relationships between niche properties and species richness across space because they have been used to study macroecological patterns in species richness, including studies that link richness to niche properties [18,21,30]. Among the Anurans, treefrogs (family Hylidae) exhibit wide variation in species richness across the Americas and thus lend themselves to the study of how regional species richness is associated with niche properties. This family is the most diverse in the order Anura, with approximately 1,081 species [34], primarily distributed in the Americas, representing 70% of the group's diversity. Hylidae exhibits a strong LDG pattern [35] with a peak of species richness in South America, a region suggested as the probable ancestral area for this clade [27]. A negative correlation between species richness and both temperature and precipitation breadth was found for North American treefrogs [18]. However, it remains unknown whether this pattern holds across the entire latitudinal distribution, including tropical clades.

In this study, we evaluate three hypotheses about the relationship between niche properties (breadth, marginality, and position) and treefrog species richness using the unidimensional approach (classic method) and a multivariate climatic approach: the niche- breadth, marginality, and phylogenetic niche position hypotheses. The first approach estimates properties for each niche dimension independently, such as temperature and precipitation. The second approach incorporates various niche dimensions into a multivariate (n-dimensional) space, which underlies all niche properties. We expect species richness to decrease as a function of the mean niche breadth, marginality, and position of all the co-distributed treefrog species (Table 1).

**Table 1. Niche properties and the expected relationship with species richness.**

| Niche property | Relationship with species richness | References |
|---|---|---|
| Breadth | Areas with higher species richness (i.e., the tropics) should contain species with narrower niche breadth. | [15–17] |
| Marginality | Areas with higher species richness should contain species with low marginality (niche centroid close to average climate conditions). | [20–24] |
| Phylogenetic Position | Areas with higher species richness should contain species with low phylogenetic niche position (i.e., species whose niches are more similar to those of the most recent common ancestor). | [26,27,29] |

## Materials and methods

### Species occurrences and environmental data

We used the AmphibiaWeb database [34] to obtain a list of 703 species of treefrog distributed in the Americas. We then obtained occurrence records for each species through the GBIF portal using the 'rgbif' package [36]. The records for each species were individually reviewed, focusing on two factors: i) taxonomic changes according to AmphibiaWeb, and (ii) geographically atypical records (i.e., records out of the known range of the species). Next, we used the 'bdc' package [37] to perform spatial filters, such as records with low accuracy (i.e., less than two decimal positions) in coordinates and records in the ocean. We applied a spatial threshold of five kilometers among occurrence records to reduce spatial auto-correlation and retain a single occurrence per pixel. We chose this threshold to match the resolution of our environmental data. Once we obtained a database of manually curated records, we used it to delimit an area within the Americas where at least one treefrog species could be present. Because treefrogs are not distributed in extreme latitudes (e.g., northern Canada, southern Argentina and Chile), the analysis area was defined using all species for which records of occurrence were obtained. This area was delimited by selecting the Tropical Nature Conservancy terrestrial ecoregions that contain at least one hylid species [38].

We obtained the climatic variables from WorldClim v.2.1 [39] at a resolution of 2.5 min (~5 km). These variables represent general tendencies and annual temporal variation in temperature and precipitation regimes established over a ~ 30-year interval. We omitted four bioclimatic variables (bio 8, bio 9, bio 18, bio 19) as they present outliers mainly in the tropics [40]. The remaining 15 variables were masked to the study area and transformed into three principal component raster layers using the 'kuenm' R package [41]. This allowed us to reduce the dimensionality and collinearity of the climate data to three axes for the modeling algorithm and make niche properties comparable because they are quantified in the same environmental space.

### Ecological niche modeling and species richness

We generated niche models using the minimum volume ellipsoids (MVE) algorithm in the 'ellipsenm' R package [42]. This method assumes that the fundamental niches are convex and have an ellipsoidal shape when multiple dimensions are considered, consistent with Hutchinson's niche definition [43]. Ellipsoids are considered a suitable method for modeling ecological niches and have been widely implemented within this discipline in recent years [44–46].

We created ellipsoids for each species using 95% of the occurrences (5% omission). Then, to obtain a species distribution, we projected each ellipsoid to the entire geographic area of analysis. A species was considered present in all geographic areas whose climates fell within the ellipsoid, and absent in areas entirely outside the ellipsoid. To mitigate commission errors produced by no geographic restrictions in niche models, we constrained them by creating a minimum convex polygon using species occurrences and then adding a 25 km buffer to account for the limited available data and the species' dispersal abilities [47]. Subsequently, distribution models were transformed into polygons and projected using the Mollweide equal area projection with the 'terra' package to avoid the area and distance deformation effect due to the planet's curvature. Then, species richness across the study region was estimated by quantifying the number of overlapping polygons (species) within grid cells of one-degree resolution (~110 km). This procedure was done using the *lets.presab* function of the 'letsR' package [48] which generated a presence-absence matrix (PAM) of species within grid cells and from which species richness values and average niche properties (see below) were obtained. Because we could not collect enough occurrence data to model all hylid species that occur in the region, we verified that the richness pattern obtained from our models (n = 441 species) did not differ from that obtained from all expert range maps from IUCN (n = 711 species) [49]. We correlated both richness estimates using a Spearman correlation test (data did not meet normality assumptions). Ellipsoid models exhibited consistent patterns (i.e., rho > 0.8) relative to those derived from IUCN range maps. Subsequent analyses were conducted based only on the ellipsoid models.

## Treefrog climatic niche properties

We calculated each niche property separately using the unidimensional and multidimensional approaches. Even though the niche is a multidimensional concept [10], we are still far from understanding the interactions and relative importance of each niche dimension in macroecological patterns [50,51]. Therefore, exploring both unidimensional and multidimensional approaches allows us a more comprehensive interpretation of our results. Further, this enabled a comparison of the results obtained with those of previous studies.

To measure niche breadth in the multidimensional approach, we used the ellipsoid volume of each species with the 'ellipsenm' package [42], while for the unidimensional approach, we measured niche breadth as the range of each variable. For temperature, niche breadth was calculated as the difference between the minimum temperature value of the coldest month (bio6) and the maximum value of the warmest month (bio5). For precipitation, we used the difference between the minimum precipitation value of the driest month (bio14) and the maximum value of the wettest month (bio13).

To measure niche marginality, we first estimated the centroid for each species and then measured the distance to the average environmental conditions in the study area. For the multidimensional approach, each species' centroids were estimated in the modeling process as an x, y, z coordinate in the environmental space estimated through the first three principal components. Niche marginality was estimated as the Mahalanobis distance between the niche centroid and the mean value of the principal components (environments) estimated in the study area, that is, the point in multivariate space where the components have a value of zero (pc1 = 0, pc2 = 0, pc3 = 0). In contrast, for the unidimensional approach, we calculated the centroid of each niche dimension (i.e., temperature and precipitation) as the middle value between the minimum and maximum values of temperature and precipitation used to find their breadth. Then, to estimate marginality in each dimension, we calculated the Euclidean distance between each species' centroid and the mean value of mean annual temperature (bio1 = 11.4°C) and annual precipitation in the study area (bio12 = 1029.7 mm).

To measure the phylogenetic niche position of each species, we first had to infer the niche of the most recent common ancestor of the clade. To do this, we performed an ancestral state reconstruction analysis based on the niche centroid values of the current species based on the multidimensional and unidimensional approaches using the 'ape' and 'geiger' R packages [52,53]. We first selected the clade corresponding to the family Hylidae from the most recent anuran phylogeny [54], which includes 69% of the total anuran diversity and maintains the same taxonomy as AmphibiaWeb. Next, we pruned the tree to include only those species for which niche traits could be estimated and that were represented in the phylogeny (n = 369). We reconstructed the ancestral states of the niche centroids based on this phylogenetic hypothesis and the niche centroids estimated under the univariate and multivariate approaches for the current species. We tested four evolutionary models: Brownian motion, Ornstein-Uhlenbeck, Early Burst, and Rate Trend using the *fitcontinous* function of the 'geiger' package [52]. To evaluate which model best described the evolution of centroids, we compared them using the Akaike Information Criterion (AIC) and AIC weights. Once the best evolutionary model was selected, we performed ancestral reconstructions for each centroid (principal components, temperature, and precipitation) using the *reconstruct* function from the 'ape' package [53], including the parameters from the selected model. We used the reconstructed value of each component as the ancestral niche centroid. Then, we used the 'ellipsenm' package [42] to calculate the Mahalanobis and Euclidean distances from the niche centroid of each species to the ancestral centroid under the multidimensional and unidimensional approaches, respectively. To represent niche properties in geography, we first associated each species' niche property with its geographic distribution (polygon) and then used the *lets.maplizer* function from 'letsR' package [48] to rasterize them. Finally, we created bivariate maps of each niche property and species richness in ArcGIS Pro v. 3.5 [55].

## Statistical analysis

To evaluate the three hypotheses about the spatial pattern of treefrog richness, we performed simultaneous autoregressive models (SAR) using the *errorsarlm* function from the 'spatialreg' R package [56]. We evaluated correlations among predictor variables using Spearman's rank correlation and avoided the simultaneous inclusion of correlated variables

(>0.7). For both niche properties estimation approaches, we tested three spatial weight matrix schemes (row standardized, globally standardized, and variance stabilizing) and two distance classes (minimum and maximum) as suggested in previous studies [57]. We selected the best model based on the AIC.

We used two methods to discard the possibility that our results could be artifacts of geographical limits or co-distribution patterns. The first method was the Geometric Constraint Model, which includes a mechanistic hypothesis in which richness patterns emerge solely from geometric constraints on species ranges. This method randomizes each species distribution, and then the species richness and average niche properties are recalculated to evaluate if the same observed relationship is recovered. We created 100 random PAMs, hence random richness patterns, maintaining the number of species and their range size constant, but at random positions. This null model tests if the observed estimates are influenced by a structural bias in aggregated species niche properties driven by repeated species co-occurrences [58]. Our null hypothesis was based on cohesive range models using the spreading-dye model [59], where for each species, an initial site within the domain is randomly selected, followed by the addition of neighboring cells until the number of occupied grid cells for the focal species is reached. The second method was a null model, in which we evaluated whether the estimates explaining the observed richness differ from those obtained under a random distribution of mean niche properties in geography. In this null model, we retained the geographic distributions of the species but randomized 100 times their niche properties among the ranges to estimate the geographic patterns of each property subsequently. In both null models, we tested whether the magnitude (slope) of the SAR models differed from the slopes derived from relating richness and niche properties under the null models.

## Results

### Species occurrences and environmental data

We gathered occurrences for 636 treefrog species distributed across the Americas. Of these, 441 species (~70%) fulfilled the minimum requirement of five unique localities necessary for niche modeling using minimum volume ellipsoids. Principal component analysis (PCA) of bioclimatic variables revealed that the first three principal components accounted for 91% of the total variance. Temperature variables predominantly influenced the first component, whereas the second and third components were mainly associated with precipitation variables (S1 Table).

### Ecological niche modeling and species richness

The spatial pattern of species richness recovered using our niche models was congruent with the pattern revealed when using all IUCN species maps, showing high diversity in the Neotropical region and lower diversity in temperate zones, with the Atlantic Forest, Amazon, and Guiana Shield standing out as richness hotspots (Fig 1). Further, we found a positive, strong correlation between richness values obtained from ellipsoid models and IUCN polygons ($rho = 0.96$, $p = 2.2e-16$). Although both patterns exhibited overall consistency, disparities in richness values were observed, particularly in the northern Atlantic Forest. This was due to the unavailability of presence points for certain species for which IUCN polygons are available (S1 Fig).

### Treefrog climatic niche properties

The following genera had at least one or multiple niche properties with minimum or maximum values out of all species considered: *Boana*, *Dendropsophus*, *Smilisca*, *Scinax*, *Ptychohyla* and *Pseudacris*. However, there was no consistency in the identity of species with the minimum or maximum values between approaches (unidimensional versus multidimensional) or niche dimensions (temperature versus precipitation; S1 File). In other words, the species with the widest multivariate niche breadth was not necessarily the one with the largest temperature or precipitation breadth, and this applies to all niche properties. Further, for the estimation of niche properties using ellipsoids, we found a tendency towards low

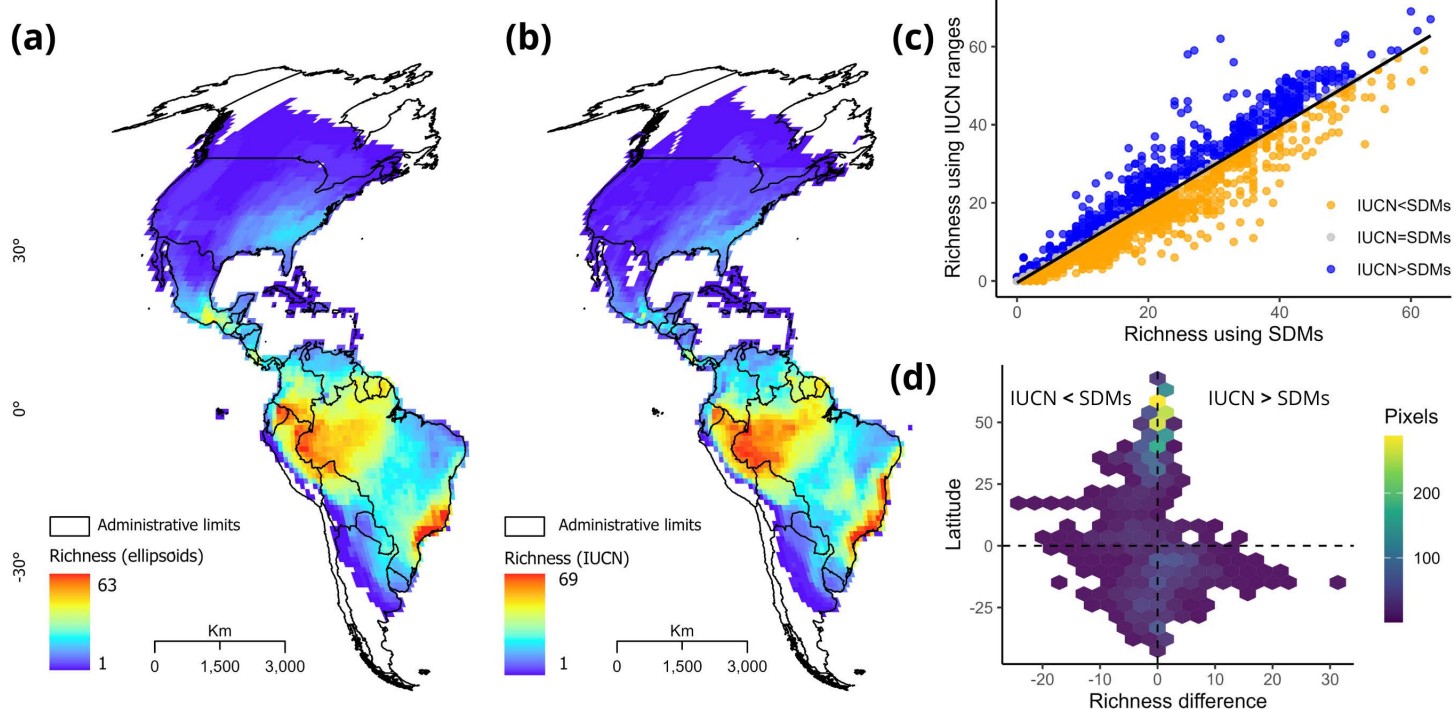

**Fig 1. Geographical species richness patterns of treefrogs (Mollweide projection) obtained from SDMs using the minimum volume ellipsoid algorithm (a) and IUCN distribution polygons (b); their congruence (c) and differences through latitude (d).** In panel (d), positive richness differences indicate cases where IUCN ranges yield higher richness than SDMs. By contrast, negative differences indicate cases with higher richness values derived from SDMs than IUCN. The frequency of each difference in geography is represented by the colors. Administrative country limits are reprinted from GADM version 4.1 under a CC BY license, with permission from https://gadm.org/license.html, original copyright 2018.

values of breadth, marginality, and position for most of the species in the analysis. In contrast, the univariate approach exhibited a more heterogeneous distribution of niche properties values (S1 File).

The geographic distribution of mean niche breadth differed between the multidimensional and unidimensional approaches. For the multidimensional approach, the niche breadth (ellipsoid volume) was wider in the tropics and narrower in temperate zones (Fig 2a). By contrast, temperature breadth was narrower in tropical regions and wider in the northern temperate zone (Fig 2b). The geographic distribution of precipitation breadths mirrored the pattern obtained from the multivariate approach (Fig 2c).

The most marginal multidimensional niches were found in the Pacific region and eastern slope of the Andes in Ecuador, characterized by high richness of treefrogs (Fig 3a). Using only temperature, most of lowland South America is characterized by high average marginality (Fig 3b), coinciding with most areas of high species richness. On the other hand, precipitation showed the opposite pattern, with low average marginality in areas with high species richness, consistent with our expectations (Fig 3c).

For the phylogenetic niche position, the evolutionary model that best fits the evolution of niche position using both approaches was the Ornstein-Uhlenbeck model (S2 Table). For the multidimensional approach, the ancestral reconstruction suggests that the treefrog's ancestor centroid was at position pc1: 2.99, pc2: 0.85, pc3: 0.366. Further, we found that the areas in the Americas that are near to the ancestral niche (i.e., 5% percentile) occur in the Neotropical region and are associated to a mean annual temperature of 25°C and a mean annual precipitation of 2000 mm (S2 File). For the unidimensional approach, ancestral centroids were reconstructed at 19.9°C and 213.2 mm for annual monthly temperature and

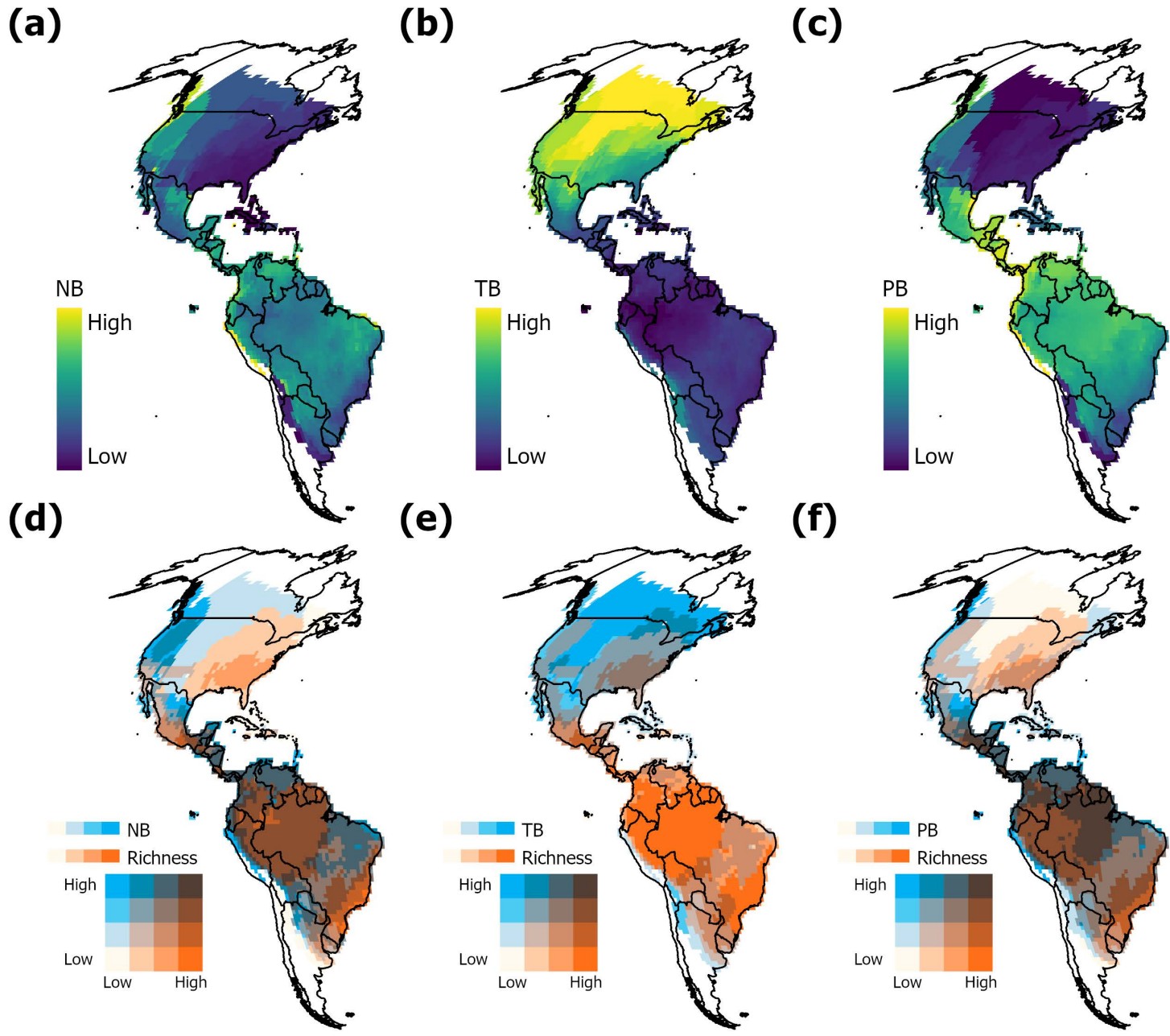

**Fig 2. Geographical pattern (Mollweide projection) of niche breadth and species richness.** (a) Niche breadth (NB) using MVE; (b) temperature breadth (TB); (c) precipitation breadth (PB). Panels d to f correspond to bivariate maps between amplitude properties and species richness. (d) Richness-NB; (e) Richness-TB; (f) Richness-PB. Administrative country limits are reprinted from GADM version 4.1 under a CC BY license, with permission from https://gadm.org/license.html, original copyright 2018.

precipitation respectively. In general, distribution patterns for niche position were consistent between unidimensional and multidimensional approaches, indicating average low position values in the Neotropical region, with the highest species richness (Fig 4).

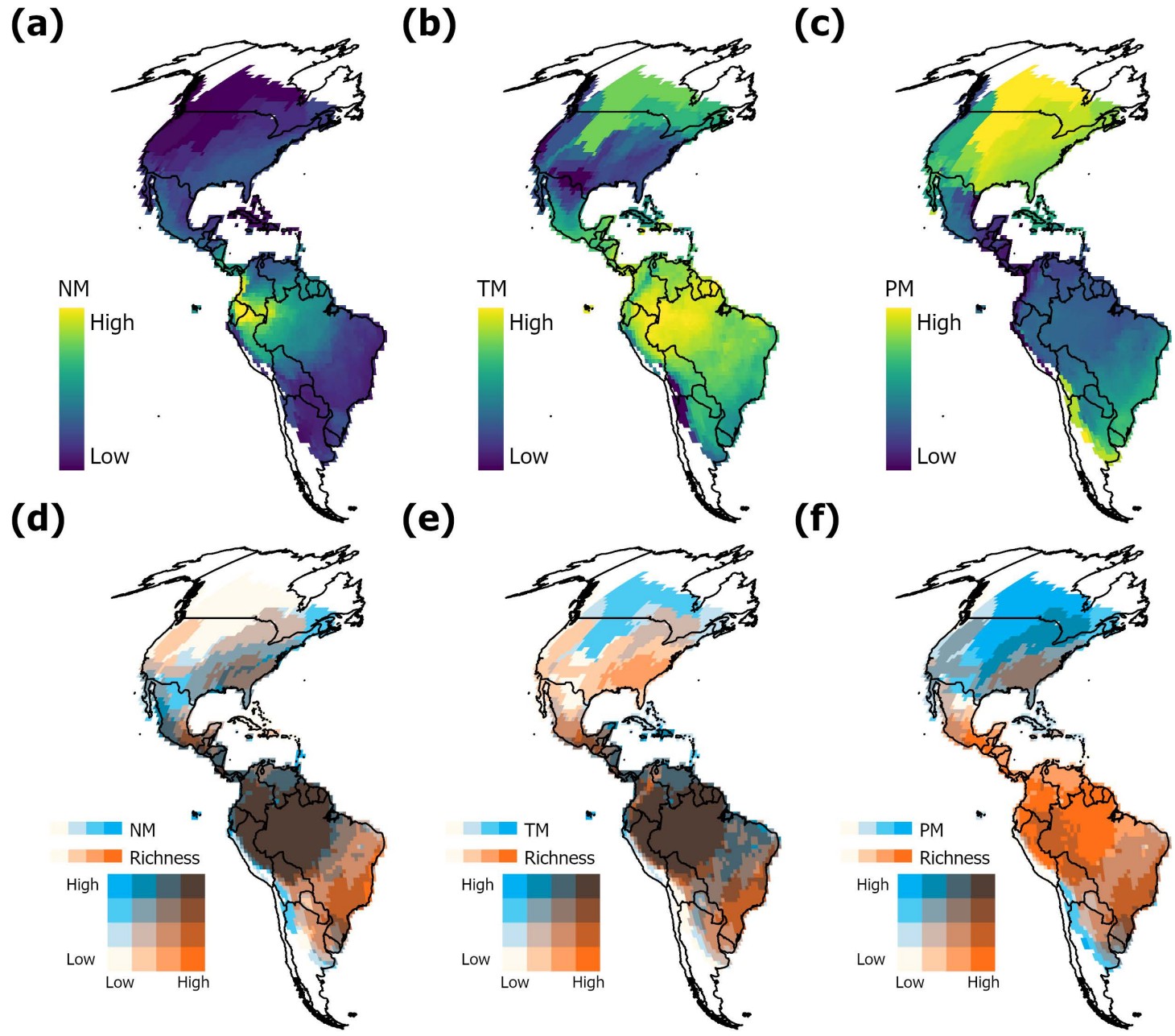

**Fig 3. Geographical pattern (Mollweide projection) of marginality and species richness.** (a) Niche marginality (NM) using MVE; (b) temperature marginality (TM); (c) precipitation marginality (PM). Panes d to f correspond to bivariate maps between marginality properties and species richness. (d) Richness-NM; (e) Richness-TM; (f) Richness-PM. Administrative country limits are reprinted from GADM version 4.1 under a CC BY license, with permission from https://gadm.org/license.html, original copyright 2018.

## Statistical analysis

The SAR models recovered the expected negative relationship between richness and niche breadth, irrespective of dimensionality, but results were variable for position and counter to our expectations for marginality (Table 2). We ran two separate models for the unidimensional approach based on precipitation because niche breadth and marginality were

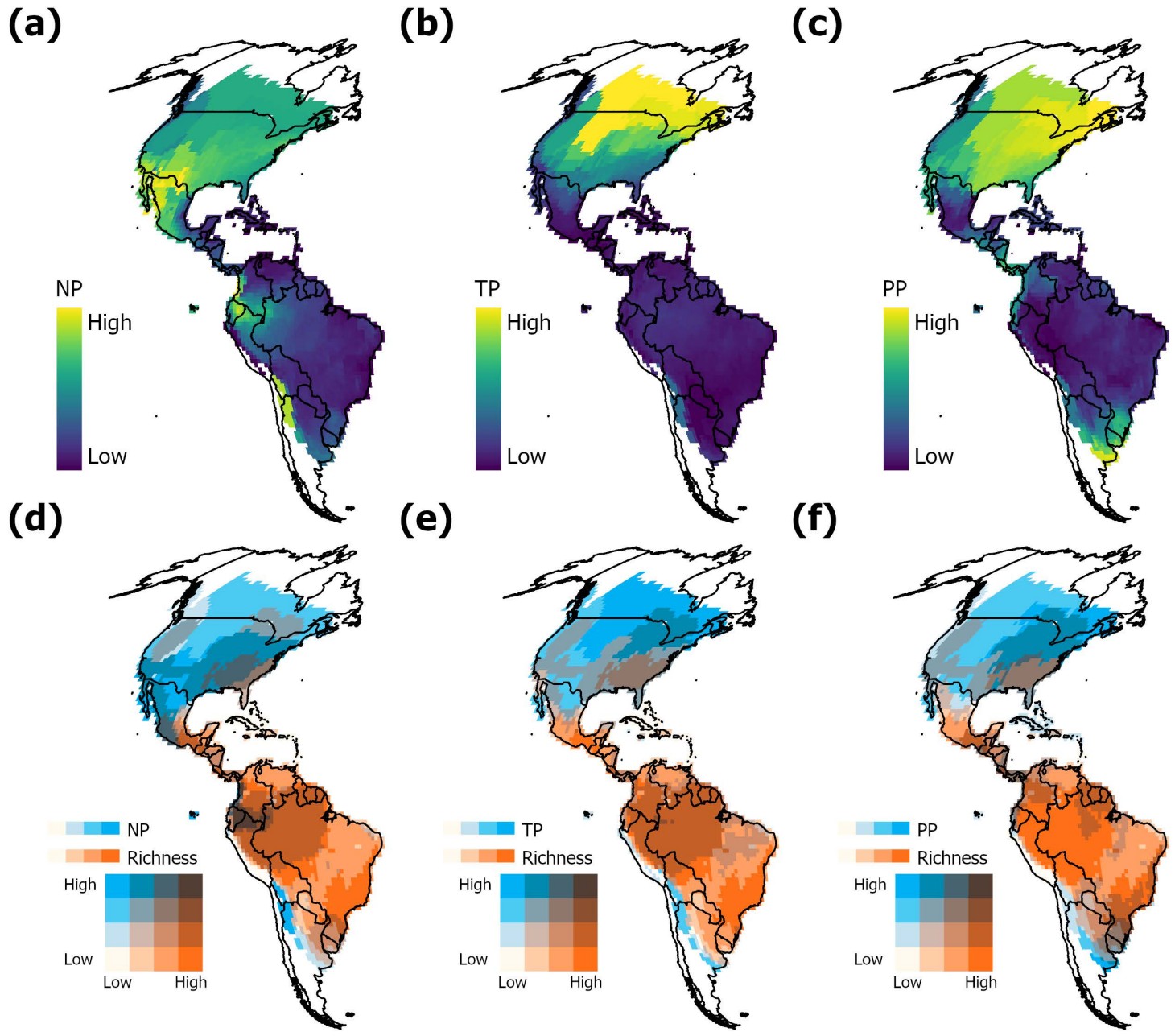

**Fig 4. Geographical pattern (Mollweide projection) of phylogenetic position and species richness.** (a) Niche position (NP) using MVE; (b) temperature position (TP); (c) precipitation position (PP). Panes d to f correspond to bivariate maps between position properties and species richness. (d) Richness-NP; (e) Richness-TP; (f) Richness-PP. Administrative country limits are reprinted from GADM version 4.1 under a CC BY license, with permission from https://gadm.org/license.html, original copyright 2018.

highly correlated (*rho* = −0.99, *p* = 2.2e-16). Sites with species whose centroids were closer to the ancestral niche had higher richness with the multidimensional approach and precipitation but not with temperature, where the opposite relationship was found. The SAR models' evaluation and spatial autocorrelation can be found in the Supporting information (S3 File).

**Table 2. Simultaneous autoregressive models of species richness and niche properties under both estimation approaches.** All models followed a variance stabilizing weight matrix (S) using the minimum distance. NB = Niche breadth, NM = Niche marginality, NP = Niche position, TB = Temperature breadth, TM = Temperature marginality, TP = Temperature position, PB = Precipitation breadth, PM = Precipitation marginality, PP = Precipitation position.

| Approach | Model | Estimate | p-value |
|---|---|---|---|
| Multidimensional | Richness ~ NB + NM + NP | NB = −1.7 | <0.01 |
| | | NM = 3.2 | <0.01 |
| | | NP = −2.8 | <0.01 |
| Unidimensional | Richness ~ TB + TM + TP | TB = −3.1 | <0.01 |
| | | TM = 0.5 | 0.007 |
| | | TP = 1.1 | 0.003 |
| | Richness ~ PB + PP | PB = −0.6 | 0.0008 |
| | | PP = −0.8 | <0.01 |
| | Richness ~ PM + PP | PM = 0.4 | 0.017 |
| | | PP = −0.8 | <0.01 |

Results indicate that the relationships established between richness and niche properties cannot be generated by the Geometric Constraint Model or by a random distribution of average niche properties for the multidimensional approach and only for temperature breadth under the unidimensional approach (S4 File). The established relationships with properties estimated using the unidimensional approach, such as temperature and precipitation marginality, temperature and precipitation position, should be interpreted with caution because the same relationships can be generated under one or both methods.

## Discussion

We tested three hypotheses about the relationship between treefrog species richness in the Americas and the climatic niche attributes of cooccurring species: breadth, marginality, and position. Our results provide support for two out of the three general expectations from these hypotheses (Table 1). Sites with high treefrog species richness in the Americas tend to be composed of species with narrower niches and similar to that of their common ancestor (i.e., low phylogenetic niche position) under the multivariate approach and for precipitation under the univariate approach. Results from the phylogenetic niche position analysis support the tropical niche conservatism hypothesis for the American treefrogs (except for temperature position). Therefore, sites with high species richness contain species with niches similar to their ancestral niche and provide clues as to the climatic conditions related to sites of origin. In addition, our results support the notion that more species can be packed if their niches are narrower, in line with Janzen´s hypothesis for temperature and MacArthur's niche packing hypothesis [15,60]. We found a positive relationship between niche marginality and richness, an outcome opposite to our expectation (Table 1), which we believe is related to the way this niche metric is calculated. The niche marginality property assumes that the most frequent environments in geography should be well represented by the mean. In our study area, the distribution of environments is bimodal (S2 Fig), and thus the interpretation of the metric is not straightforward. Overall, we conclude that to establish patterns of species richness distribution, it is not only useful to have a good comprehension of the geographic distribution of climate, but also about physiological tolerances, limitations, and preferences that organisms have evolved throughout their history.

The richness pattern obtained from the MVEs was consistent with previous reconstructions for the family [27] and with that estimated based on the IUCN polygons. However, we observed differences in the Amazonian region, where sampling biases are common [61]. Although these differences do not exceed 3.1% of total of cells, the variation in the resulting species assemblages could potentially impact the subsequent estimation of the average niche property in cells [62]. However, it seems reasonable to assume that, as the overall richness pattern remains consistent between SDMs and expert ranges,

the spatial pattern of niche properties should remain conserved (i.e., narrower niche breadth in tropics than in temperate regions); still, this has not been formally evaluated.

Climatic niche breadth was the only property that consistently exhibited a negative relationship with richness, whether it was based solely on temperature, precipitation, or on a multivariate representation of climate. The geographic distribution of the average niche breadth using ellipsoid volumes was similar to the distribution of mean precipitation breadth, which may indicate that most variation in niche volume is due to precipitation. From an eco-physiological perspective, precipitation is presumed to play a key role for amphibians, as it is linked to water availability and environmental humidity, which is essential for their reproduction, abundance, and survival [63]. SAR models indicated that niche breadth is negatively associated with species richness, nonetheless, the highest effect size on species richness was related to temperature breadth (Table 2). This corroborates the hypotheses proposed by Janzen [15] and Currie [16] of an association between high species richness and narrower climatic niches in tropical regions and is consistent with previous findings [8,18,64]. Interestingly, despite tropical latitudes exhibiting higher variation in precipitation than temperate regions [17], the sites with the highest variation in precipitation (e.g., Chocó-Darién, Western Ecuador and Magdalena-Urabá moist forests, Sinú Valley Dry Forests and Guajira-Barranquilla xeric scrub ecoregions) do not present high treefrog richness (Fig 2c, 2f).

Climatic niche marginality was positively related to species richness. Sites with higher species richness were composed of species with high niche marginality for temperature and precipitation. This suggests that most species occur under conditions distant from average temperature and precipitation in the study area (11.4°C and 1029.7 mm; S2 Fig). We identify two problems with niche marginality. First, average conditions are subject to the study area. Second, the distribution of climatic variables in geography is frequently not normal, hence the average does not necessarily represent the most frequent conditions. In our study area, the average temperature (11.4 °C) differs markedly from the most frequent temperature value (~26°C), whereas the precipitation average (1029.7 mm) is comparatively close to the most prevalent precipitation value (~850 mm). Although useful as a reference measure, niche marginality is not a property that depends solely on the species; it also depends on the delimited study area and thus could be highly subjective.

Our results concerning phylogenetic niche position support the tropical niche conservatism hypothesis since sites with high richness were composed of species whose niches were closer to the ancestral niche at tropical latitudes with mesic climates (20°C monthly temperature and 213.2 mm monthly precipitation). This agrees with previous research on treefrogs that estimated the conditions of the ancestral niche around 18.3°C and 1745 mm year$^{-1}$ [18,27], and for other anuran groups such as ranids [65]. These conditions are colder and less humid than those typical of the tropical climates associated with lowland Amazonia and Chocó [66]. When temperature alone was assessed, the relationship between richness and temperature position was positive, indicating higher richness at sites with species whose niches are distant from that of the ancestor. This is an interesting result suggesting that treefrogs may have more evolutionary lability in their temperature physiological tolerance than in their precipitation tolerance [67]. Different niche axes may exhibit different evolutionary rates, although the few available studies have not supported any clear relationship between niche evolution rate and niche properties [68]. SAR models partially corroborate these results when all properties are included simultaneously to explain the richness pattern (Table 2).

Comparison with the Geometric Constraint models and the null models that distributed niche properties randomly across species ranges revealed that the spatial variation in treefrog species richness differed significantly from geometrical constraints or null expectations under the multidimensional approach. In contrast, when using the univariate approach, deviations from geometric expectations were detected only for temperature breadth. These suggest that multidimensional properties, as well as temperature breadth, may explain species' richness in American treefrogs, beyond geometric constraints. Under the unidimensional approach, we found that only breadth and position captured non-random processes that shape treefrog richness patterns, while marginality may play a more context-dependent role, as we described before. Therefore, niche conservatism restricts species dispersal to temperate areas, accumulating more species in the tropics over time, evolving narrower niche breadths [15,29,60].

Using individual niche dimensions can provide valuable insight into the role of each one in explaining macroecological patterns and evolutionary processes. These dimensions represent the potential adaptive responses to changes in temperature or precipitation through time, which allow niche evolution and diversification [7,14]. However, the niche is a multivariate property of the species, with heterogeneity in the evolutionary lability of each axis, determining the limitations and evolutionary potential of each species in the face of climate change [69]. For example, species diversification might be more likely along a particular climatic niche dimension [70]. The number of niche dimensions in which niche properties vary influences species divergence, convergence, and coexistence, ultimately determining species richness [71]. Interpretations of patterns or processes under the niche concept are more appropriate under a multidimensional perspective, which provides an overview of the species' niche rather than a single dimension. However, the multidimensional approach may obscure certain patterns if a single niche dimension accounts for most of the variation underlying the estimation of niche properties, or if different dimensions respond in an opposite manner. Therefore, both approaches (i.e., multivariate and univariate) serve as complementary methodologies that allow better interpretation of the relationships between niche properties and biogeographical patterns, such as the latitudinal diversity gradient.

Finally, we identify caveats for niche properties that should be considered in future studies. First, niche marginality and phylogenetic niche position depend on methodological decisions, including selecting the area of analysis for marginality or defining the phylogenetic scale of analysis for niche position [72]. These factors may contribute to the variability in estimates of these properties, which can have important consequences in a macroevolutionary context, where the rates of evolution of these properties or their relationships with species diversification rates are of particular interest [64,73]. For example, Meyer and Pie [21] successfully used climatic prevalence (the frequency of a climatic category in geography) as a predictor of species richness. Nonetheless, the prevalence, as the marginality, is subject to the delimitation of the area of study. According to the marginality definition proposed by Hirzel [20], this area is essential for estimating the average available environments. However, there is ample debate over how such area is delimited [47], and this may result in drastic variation in the marginalities estimated.

Understanding geographic patterns of species richness remains an exciting area in macroecology. Although our study emphasizes the relationship between richness and niche properties, future studies should also embrace how these properties evolve, and their effects on diversification and dispersal rates [74].

## Supporting information

**S1 Table. Results of Principal Component Analysis.** Variable loadings for the three principal components.
(DOCX)

**S2 Table. Ancestral reconstruction of niche centroids.** Statistical fit and comparison of four evolutionary models of ancestral state reconstruction.
(DOCX)

**S1 Fig. Location of non-modeled species.** Distribution and concentration of species with less than five occurrences.
(DOCX)

**S2 Fig. Distribution of temperature and precipitation values in the Americas.** Geographic prevalence of temperature and precipitation values in the study area.
(DOCX)

**S1 File. Climatic niche properties of American treefrogs.** Ranking and distribution of species' niche properties.
(DOCX)

**S2 File. Areas and climatic conditions close to the ancestral position using the multivariate approach.** Geographic areas with similar environments to the ancestral niche position and distribution of environments in the study area in relation to the ancestral niche position.
(DOCX)

**S3 File. Spatial autocorrelation and evaluation of SAR models.** Statistical fit of various spatial autocorrelation models on climatic niche properties.
(DOCX)

**S4 File. Geometric constraint and Null models results.** Distribution of estimated slopes between richness and climatic niche properties under two null models (geometric constraints and random distribution of mean niche properties).
(DOCX)

## Acknowledgments

We thank Daniel Valencia for the friendly review of this manuscript. Likewise, the members of the Laboratorio de Bioclimatología and the Grupo de Ecología y Evolución de Vertebrados provided their support and comments to improve this work.

## Author contributions

**Conceptualization:** Felipe A. Toro-Cardona, Julián A. Velasco, Jesús Pinto-Ledezma, Sean M. Rovito, Fabricio Villalobos, Octavio R. Rojas-Soto, Juan L. Parra.

**Data curation:** Felipe A. Toro-Cardona, Juan L. Parra.

**Formal analysis:** Felipe A. Toro-Cardona, Julián A. Velasco, Jesús Pinto-Ledezma, Sean M. Rovito, Fabricio Villalobos, Octavio R. Rojas-Soto, Juan L. Parra.

**Investigation:** Felipe A. Toro-Cardona, Octavio R. Rojas-Soto, Juan L. Parra.

**Methodology:** Felipe A. Toro-Cardona, Julián A. Velasco, Jesús Pinto-Ledezma, Sean M. Rovito, Fabricio Villalobos, Octavio R. Rojas-Soto, Juan L. Parra.

**Resources:** Julián A. Velasco, Juan L. Parra.

**Supervision:** Julián A. Velasco, Jesús Pinto-Ledezma, Sean M. Rovito, Fabricio Villalobos, Octavio R. Rojas-Soto, Juan L. Parra.

**Validation:** Felipe A. Toro-Cardona, Juan L. Parra.

**Visualization:** Felipe A. Toro-Cardona, Octavio R. Rojas-Soto, Juan L. Parra.

**Writing – original draft:** Felipe A. Toro-Cardona, Juan L. Parra.

**Writing – review & editing:** Felipe A. Toro-Cardona, Julián A. Velasco, Jesús Pinto-Ledezma, Sean M. Rovito, Fabricio Villalobos, Octavio R. Rojas-Soto, Juan L. Parra.

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
