## [Decision Letter · Decision Letter 0]

27 Nov 2025

PONE-D-25-34069Climatic niche properties shape treefrog diversityPLOS ONE

Dear Dr. Parra,

Thank you for submitting your manuscript to PLOS ONE. After careful consideration, we feel that it has merit but does not fully meet PLOS ONE’s publication criteria as it currently stands. Therefore, we invite you to submit a revised version of the manuscript that addresses the points raised during the review process.

We look forward to receiving your revised manuscript.

Kind regards,

Daniel de Paiva Silva, Ph.D.

Academic Editor

PLOS ONE

Journal Requirements:

“Felipe A. Toro-Cardona received funding from the Posgraduate Direction at Universidad de Antioquia for international travel.”

6. We note that Figures 1, 2, 3, 4, S2 and S6  in your submission contain map images which may be copyrighted. All PLOS content is published under the Creative Commons Attribution License (CC BY 4.0), which means that the manuscript, images, and Supporting Information files will be freely available online, and any third party is permitted to access, download, copy, distribute, and use these materials in any way, even commercially, with proper attribution. For these reasons, we cannot publish previously copyrighted maps or satellite images created using proprietary data, such as Google software (Google Maps, Street View, and Earth). For more information, see our copyright guidelines: http://journals.plos.org/plosone/s/licenses-and-copyright.

1. You may seek permission from the original copyright holder of Figure(s) [#] to publish the content specifically under the CC BY 4.0 license.

Additional Editor Comments:

Dear Dr. Parra,

After this first and thorough review round, three reviewers indicated the need for improvements in your manuscript. I acknowledge the time you needed to wait for us to gather all three reviews. The editorial practice is very difficult since the COVID-19 pandemic, and finding available and reliable reviewers has been challenging for journal editors. Therefore, I would like to thank you and your co-authors for your patience and the reviewers for their partnership.

Regarding you You will see that there are practically all kinds of decisions available. Still, all of them were positive from my point of view. Please see that along with text improvements suggested by all three reviewers, one of them made significant suggestions of improvement regarding your methods. Please take special attention to all suggestions, but specially those from this specific reviewer.

I hope the provided suggestions allow you and your co-authors to improve your text and resubmit it and have it accepted for publication in PLoS One soon.

Sincerely,

Daniel Silva

Reviewers' comments:

Reviewer's Responses to Questions

**Comments to the Author**

1. Is the manuscript technically sound, and do the data support the conclusions?

Reviewer #1: Partly

Reviewer #2: Yes

Reviewer #3: Partly

2. Has the statistical analysis been performed appropriately and rigorously? 

Reviewer #1: Yes

Reviewer #2: Yes

Reviewer #3: Yes

3. Have the authors made all data underlying the findings in their manuscript fully available?

Reviewer #1: Yes

Reviewer #2: Yes

Reviewer #3: Yes

4. Is the manuscript presented in an intelligible fashion and written in standard English?

Reviewer #1: No

Reviewer #2: Yes

Reviewer #3: Yes

5. Review Comments to the Author

Reviewer #1: Comments to the authors:

This manuscript presents a compelling macroecological investigation into the latitudinal diversity gradient (LDG) of American treefrogs, using niche properties (breadth, marginality, and position) to explore hypotheses linking species-climate interactions to patterns of species richness. The study addresses an important topic and represents a valuable contribution to the field. However, to strengthen the manuscript, several areas require attention, particularly regarding methodological clarity, the consistency of the narrative, and the necessity of certain analyses. My specific comments below are offered to help the authors improve the rigor and impact of their work.

The manuscript would benefit from greater consistency in terminology. The core concepts are defined as niche breadth, marginality, and position. However, the text frequently introduces other theoretical names like "species-area" and "tropical niche conservatism" without clearly linking them to these core properties. This is confusing for the reader. I strongly recommend that the authors consistently use the terms "breadth," "marginality," and "position" throughout, and carefully justify how other hypotheses relate to these specific metrics.

In the abstract the phrase between lines 37-39 is confusing, is “precipitation” a single hypothesis?

In the abstract the phrase between the lines 43-46 is also confusing. Please rephrase it.

Upon first mentioning the latitudinal diversity gradient (LDG) in line 50, the manuscript should briefly define this fundamental pattern. Please add a concise phrase to describe the LDG as the observed decrease in species richness from the tropics to the poles.

In lines 57-58, why are you using dashes?

In line 77, the statement that "the most frequent climates in geography are expected to hold more species" requires justification. Providing a sentence or two of explanation is crucial for the reader to understand the foundation of your hypothesis.

In line 88, the rationale for using the "common ancestor of the clades" in the analysis requires explicit justification.

The phrase on lines 105-106 appears to be disconnected from the main argument of the paragraph. Please either integrate it more smoothly by explaining its connection to the point being made, or consider moving it to a more appropriate location in the manuscript where it better fits the context.

Line 116, the word “and” is misplaced.

The Materials and Methods section requires significant clarification regarding the necessity of three specific analyses. Currently, their purpose is unclear, and as they are not discussed in the results, they detract from the manuscript's focus.

The MVE vs. IUCN richness comparison. The core methodology involves creating species ranges from MVE-derived niches. The rationale for comparing the resulting richness pattern to one derived from IUCN ranges is not explained. Since this comparison is not discussed, its inclusion seems unnecessary and should be justified or removed.

Multivariate vs. Univariate niche properties comparison: It is well-established that niche properties are multivariate. The justification for comparing these robust multivariate results to univariate approximations is lacking. This entire section is confusing and creates an inconsistent narrative. Without a clear purpose and discussion, its inclusion is difficult to support.

Null model analysis. The aim of the null model comparison is not clear. Furthermore, I must point out a conceptual issue: the "spreading-dye" model (or Geometric Constraint Model - GCM) is not a true null model but a mechanistic hypothesis that itself explains richness patterns through geometric constraints. The use and interpretation of this analysis require substantial clarification.

I strongly advise the authors to either (1) provide a compelling a priori justification for each of these analyses and integrate a thorough discussion of their results into the manuscript, or (2) remove them entirely. I am inclined to recommend removal, as this would significantly improve the manuscript's clarity, focus, and logical flow.

Line 135, this “five-kilometers radius” has a justification?

Line 173, “teste” is miswritten.

Lines 188-189, what are the environmental variables?

Line 191, “the environmental space” is based in what?

Lines 193-199, this reading is very confusing, please rephrase.

Line 201, “both approaches”? What approaches?

Line 206-208, I do not understand what you want to say here. Please clarify.

Line 222, the GIS software, what GIS software?

Line 243-247, again, I did not understand what do you want to say here, please rephrase.

Line 264, Atlantic Forest and Amazon are biomes, Guiana is a country. Please rephrase.

Line 281, “extreme values” what do you want to mean with “extreme”?

Line 283, again, “extreme niche properties”, what do you want to mean with “extreme”?

Line 356-358, you are repeating results in the discussion.

Line 364, please delete the parentheses.

Line 372, the parentheses are repeating the results, please delete it.

Lines 378, 430, and 433, the citations are miswritten, the citations should be presented in numbers.

Lines 391-393, they are “distant” or “close” in relation of what?

Reviewer #2: The manuscript highlights the relevance of considering multiple dimensions of the climatic niche to explain macroecological patterns, offering important insights into the evolution and distribution of treefrog diversity. Thus, this article presents a significant contribution to ecology and biogeography by interestingly investigating how climatic niche properties—breadth, marginality, and phylogenetic position—influence treefrog diversity in the Americas.

Reviewer #3: General comments

The manuscript presents important results, particularly in the comparison between unidimensional and multidimensional approaches, which provides a broader view of macroecological patterns. I recommend accepting the manuscript in PLOS ONE, after improving the writing to enhance clarity and objectivity in the presentation of results. My main criticisms are: (1) although the analyses show scientific rigor, in several sections the discussion does not seem fully aligned with the results obtained; (2) there is a lack of standardization in the use of terms (e.g., “marginality hypothesis” and “species-area hypothesis”), as well as in the order in which results are presented. Understanding improves when this order is consistent across sections; and (3) the text would be clearer if, instead of citing the hypotheses (e.g., “we found a negative relationship between niche breadth and species richness”), the authors cited the predictions (e.g., “species richness was higher where species exhibited narrower niches”). The hypotheses must be clearly described in the Materials and Methods, but the discussion becomes more objective when focused on the observed patterns. Below I provide suggestions for textual adjustments and more specific questions that need to be addressed.

Abstract

Lines 30–31: “the species-area hypothesis predicts higher richness in areas with more frequent climates.”

I suggest standardizing the wording to facilitate understanding of each hypothesis. For example: “the marginality hypothesis predicts higher richness in areas with more common climatic conditions.” Additionally, the meaning of “more frequent climates” and how this relates to marginality is unclear. Are more frequent climates located near the center of the species’ climatic distribution? Clarify and standardize terminology.

Lines 31–32: “the species-area hypothesis predicts higher richness in areas with more frequent climates.”

Likewise, maintain the term “position” when explaining the hypothesis. For example: “the position hypothesis predicts higher richness in areas where species niches are closer to those of their ancestors.” Remember this is the abstract, and detailed explanation will appear in the Introduction.

Lines 32–33: “We estimate niche properties of all American treefrog species using a univariate...”

Not all American species were analyzed, since some lacked sufficient occurrence records to build niche models.

Lines 40–41: “We found no support for the species-area hypothesis under both approaches.”

I suggest replacing with: “marginality hypothesis.”

Introduction

Lines 75–83: “Niche marginality... Hirzel et al. [20].”

This paragraph is unclear regarding the relationships among marginality, niche centroids, and regional climatic means. Under what conditions is the distance between a species’ niche centroid and the regional climatic mean smaller? Do species with low marginality exhibit high or low centroid-to-mean distance? Marginality relative to what exactly? Do species with high marginality have centroids closer to extreme deviations from the regional mean? I suggest reorganizing as follows: define what marginality measures, explain what constitutes high and low marginality, and then describe its expected relationship with richness.

Lines 82–83: “but the original concept corresponds to marginality proposed by Hirzel et al.”

What is the original concept proposed by Hirzel et al.? All essential information should be included in the text instead of requiring the reader to pause and search externally.

Lines 76–77: “In addition, the most frequent climates in geography are expected to hold more species.”

The meaning of “more frequent climates” remains unclear, as does why they are expected to harbor more species.

Lines 117–118: “and a multivariate climatic approach: the niche-breadth, species-area, and tropical niche conservatism hypotheses.”

Again, I suggest standardizing the terminology. Choose whether to refer to the hypotheses as niche breadth, marginality, and position, or as niche breadth, species geographic range (species-area), and niche conservatism. In some parts, they appear to represent different hypotheses.

Table 1:

The definition of marginality and how its numerical values indicate low marginality are still unclear.

Materials and Methods

Lines 134–136: “We applied a spatial threshold of a five-kilometer radius between records to reduce spatial autocorrelation and obtain a single occurrence per pixel.”

It is not clear how a 5-km radius ensures one occurrence per pixel. What is the pixel size? If two pixels fall within the 5-km radius, is only one selected? Explain the steps of this criterion and the pixel grid definition.

Lines 136–141: “Once we obtained... one occurrence record”.

Some sections are repetitive. I suggest the following revision:

“Once we obtained a manually curated database of occurrence records, we used it to delimit an area within the Americas where at least one treefrog species could occur. Because treefrogs are absent from extreme latitudes (e.g., northern Canada; southern Argentina and Chile), these regions were excluded. This area was delimited using the Tropical Nature Conservancy terrestrial ecoregions [39]”.

Lines 142–149: “We obtained the climatic variables... the same environmental space”.

This section belongs in the next subsection (Ecological niche modeling and species richness).

Lines 159–161: “Then, to obtain a species distribution... in all areas outside”.

Suggested revision:

“A species was considered present in all geographic areas whose climates fell within the ellipsoid, and absent in areas entirely outside the ellipsoid.”

Line 175: “we used a Spearman correlation test”.

Explain that the data did not meet normality assumptions, and therefore the Spearman correlation test was used.

Lines 175–177: “If the patterns exhibited consistency (i.e. rho > 0.8), subsequent analyses were conducted based on results from ellipsoid models”.

This sentence can be more direct and consistent with past-tense writing:

“Subsequent analyses were conducted based on the ellipsoid models that exhibited consistent patterns (i.e., rho > 0.8) relative to those derived from IUCN range maps.”

Lines 180–187: “We calculated each niche... wettest month (bio14)”.

I suggest standardizing the order of presentation, beginning with the unidimensional approach (first mentioned in Materials and Methods), followed by the multidimensional one. Apply this standardization across all sections.

Results

Lines 252–255: “Of these, 441 species (~70%) fulfilled the minimum requirement of five unique localities necessary for niche modeling using minimum volume ellipsoids, and 369 of those were present in the phylogenetic tree”.

I recommend stating the minimum occurrence requirement and resulting species counts in the Materials and Methods, and presenting results here directly for the final set of 369 analyzed species.

Lines 261–262: “We obtained the potential distribution models for 441 treefrog species, while 711 species distribution polygons were obtained from IUCN”.

This sentence should be in the Methods, as all methodological decisions belong there. The Results section should present only outcomes.

There is inconsistency in sample numbers:

– abstract: “all American hylids analyzed”;

– line 128: 703 species compiled;

– line 253: 441 species reached the minimum threshold;

– line 369: species included in the phylogeny;

– line 261: 441 models + 711 IUCN polygons, exceeding the initial list of 703 species.

Also, each analysis should rely on the same number of independent species. Shouldn't amplitude, marginality, and position analyses all use the same 369 species?

Lines 316–317: “For the multidimensional approach, the ancestral reconstruction suggests that the 316 treefrog's ancestor centroid was at position pc1: 2.99, pc2: 0.85, pc3: 0.366”, and Lines 322–325: The multidimensional approach showed that the areas in The Americas that are near to the ancestral niche (i.e. 5% percentile) were in the Neotropical region and are associated to an annual mean temperature of 25°C and an annual mean precipitation of 2000 mm (S6)”.

I suggest combining these sentences and standardizing the order of approaches.

Line 349: “temperature and precipitation position”.

According to Table 2, precipitation position (PP) was not significant.

Discussion

Lines 356–358: “Sites with high treefrog species richness in the Americas tend to be composed of species with narrower niche breadths and similar to that of their common ancestor”.

According to Table 2, the richness–temperature-breadth relationship was not significant. Therefore, the statement that richness is higher in areas with narrower niches is not supported. Is that correct?

Lines 356–359: “Sites with high treefrog species richness in the Americas tend to be composed of species with narrower niche breadths and similar to that of their common ancestor”.

Specify that the result applies only to precipitation.

Lines 359–360: “Our results support the tropical niche conservatism hypothesis for the American treefrogs and provide clues as to the climatic conditions related to sites of origin”

Spatial autocorrelation did not indicate statistical significance for the richness × temperature breadth relationship.

Line 368: “Climatic niche breadth was the only property that consistently exhibited a negative relationship with richness, whether it was based solely on temperature, precipitation, or a multivariate representation of climate”.

The statement includes negative but non-significant relationships, correct? In both the unidimensional (temperature) and multidimensional approaches. Niche breadth was supported only for precipitation in the unidimensional approach.

Lines 375–378: “SAR models indicate that niche breadth is negatively associated with species richness, nonetheless, the highest effect size on species richness was related to temperature breadth (Table 2)”.

Again, this is a relationship not different from random expectations.

Lines 380–383: “Interestingly, despite tropical latitudes exhibiting higher variation in precipitation than temperate regions [17], the sites with the highest variation in precipitation (e.g. Chocó and the Caribbean region) do not present high treefrog richness (Fig. 2c, 2f)”.

Is there any possible ecological explanation for this result?

Line 397: “niches”.

Specify that this refers to temperature niches (not precipitation).

Lines 420–422: “Interpretations of patterns or processes under the niche concept are more appropriate under a multidimensional perspective, which provides an overview of the species niche and not a single dimension”.

It is unclear whether “multidimensional perspective” refers to combining unidimensional + multidimensional analyses or referring exclusively to multivariate analyses. If referring to the latter, note that the unidimensional approach provided more informative results. A multidimensional approach can also introduce noise into interpretation. For instance, the role of marginality remains unclear given its subjective nature.

6. PLOS authors have the option to publish the peer review history of their article (what does this mean?). If published, this will include your full peer review and any attached files.

Reviewer #1: No

Reviewer #2: No

Reviewer #3: No

You may also use PLOS’s free figure tool, NAAS, to help you prepare publication quality figures: https://journals.plos.org/plosone/s/figures#loc-tools-for-figure-preparation

---

## [Author Response · Author response to Decision Letter 1]

12 Jan 2026

Please find responses to all comments in the Response to Reviewers document attached.

---

## [Decision Letter · Decision Letter 1]

3 Mar 2026

PONE-D-25-34069R1Climatic niche properties shape treefrog diversityPLOS One

Dear Dr. Parra,

Thank you for submitting your manuscript to PLOS ONE. After careful consideration, we feel that it has merit but does not fully meet PLOS ONE’s publication criteria as it currently stands. Therefore, we invite you to submit a revised version of the manuscript that addresses the points raised during the review process.

We look forward to receiving your revised manuscript.

Kind regards,

Daniel de Paiva Silva, Ph.D.

Academic Editor

PLOS One

Journal Requirements:

Reviewers' comments:

Reviewer's Responses to Questions

**Comments to the Author**

1. If the authors have adequately addressed your comments raised in a previous round of review and you feel that this manuscript is now acceptable for publication, you may indicate that here to bypass the “Comments to the Author” section, enter your conflict of interest statement in the “Confidential to Editor” section, and submit your "Accept" recommendation.

Reviewer #1: All comments have been addressed

Reviewer #2: All comments have been addressed

2. Is the manuscript technically sound, and do the data support the conclusions?

Reviewer #1: Yes

Reviewer #2: Yes

3. Has the statistical analysis been performed appropriately and rigorously? 

Reviewer #1: Yes

Reviewer #2: Yes

4. Have the authors made all data underlying the findings in their manuscript fully available?

Reviewer #1: Yes

Reviewer #2: Yes

5. Is the manuscript presented in an intelligible fashion and written in standard English?

Reviewer #1: Yes

Reviewer #2: Yes

6. Review Comments to the Author

Reviewer #1: Comments to the Authors:

I appreciate the efforts made by the authors in addressing the reviewers' previous comments and suggestions. The manuscript has improved as a result. However, a few minor issues remain that should be addressed prior to final acceptance:

Line 106: The term "system" is not appropriate here, as anurans represent a biological group rather than a system. Please revise accordingly.

Line 228: The acronym AIC is introduced here without prior definition. Please provide the full term at first mention.

Line 248: The acronym AIC is defined here; however, as noted above, this definition should be moved to its first occurrence in the text (line 228).

Line 256: The sentence begins with "These," yet only one null model has been described. Please rephrase for clarity and accuracy.

Line 393: Please remove the parenthetical phrase "see below" and instead provide the relevant information directly at this point in the text.

Thank you for your attention to these final revisions.

Reviewer #2: I thank the authors for their dedicated attention and for the careful and comprehensive implementation of the suggestions presented in the previous round of review; the changes made have fully addressed the issues raised, substantially enhancing the clarity, methodological rigor, and overall impact of the manuscript.

7. PLOS authors have the option to publish the peer review history of their article (what does this mean?). If published, this will include your full peer review and any attached files.

Reviewer #1: No

Reviewer #2: No

---

## [Author Response · Author response to Decision Letter 2]

9 Mar 2026

Response to reviewers’ comments [6 march 2026]

Reviewer #1:

I appreciate the efforts made by the authors in addressing the reviewers' previous comments and suggestions. The manuscript has improved as a result. However, a few minor issues remain that should be addressed prior to final acceptance:

Line 106: The term "system" is not appropriate here, as anurans represent a biological group rather than a system. Please revise accordingly.

R: We changed “system” by “group”.

Line 228: The acronym AIC is introduced here without prior definition. Please provide the full term at first mention.

R: Done.

Line 248: The acronym AIC is defined here; however, as noted above, this definition should be moved to its first occurrence in the text (line 228).

R: Done.

Line 256: The sentence begins with "These," yet only one null model has been described. Please rephrase for clarity and accuracy.

R: We changed “These” by “This”.

Line 393: Please remove the parenthetical phrase "see below" and instead provide the relevant information directly at this point in the text.

R: We deleted the “see below” and mentioned a general problem, which is developed later in discussion. We now include the following sentences: “The niche marginality property assumes that the most frequent environments in geography should be well represented by the mean. In our study area, the distribution of environments is bimodal (S8 Fig.), and thus the interpretation of the metric is not straightforward.”.

Reviewer #2:

I thank the authors for their dedicated attention and for the careful and comprehensive implementation of the suggestions presented in the previous round of review; the changes made have fully addressed the issues raised, substantially enhancing the clarity, methodological rigor, and overall impact of the manuscript.

R: Thanks for your comments.

---

## [Decision Letter · Decision Letter 2]

20 Apr 2026

Climatic niche properties shape treefrog diversity

PONE-D-25-34069R2

Dear Dr. Parra,

We’re pleased to inform you that your manuscript has been judged scientifically suitable for publication and will be formally accepted for publication once it meets all outstanding technical requirements.

Kind regards,

Daniel de Paiva Silva, Ph.D.

Academic Editor

PLOS One

Additional Editor Comments (optional):

Reviewers' comments:

Reviewer's Responses to Questions

**Comments to the Author**

1. If the authors have adequately addressed your comments raised in a previous round of review and you feel that this manuscript is now acceptable for publication, you may indicate that here to bypass the “Comments to the Author” section, enter your conflict of interest statement in the “Confidential to Editor” section, and submit your "Accept" recommendation.

Reviewer #1: All comments have been addressed

2. Is the manuscript technically sound, and do the data support the conclusions?

Reviewer #1: Yes

3. Has the statistical analysis been performed appropriately and rigorously? 

Reviewer #1: Yes

4. Have the authors made all data underlying the findings in their manuscript fully available?

Reviewer #1: Yes

5. Is the manuscript presented in an intelligible fashion and written in standard English?

Reviewer #1: Yes

6. Review Comments to the Author

Reviewer #1: (No Response)

7. PLOS authors have the option to publish the peer review history of their article (what does this mean?). If published, this will include your full peer review and any attached files.

Reviewer #1: No

---

## [Editor Report · Acceptance letter]

PONE-D-25-34069R2

PLOS One

Dear Dr. Parra,

I'm pleased to inform you that your manuscript has been deemed suitable for publication in PLOS One. Congratulations! Your manuscript is now being handed over to our production team.

Kind regards,

on behalf of

Dr. Daniel de Paiva Silva

Academic Editor

PLOS One